# Dentofacial mini- and microesthetics as perceived by dental students: A cross-sectional multi-site study

**Lívia Romsics**[1], **Angyalka Segatto**[2], **Kristóf Boa**[1], **Roland Becsei**[1], **Noémi Rózsa**[3], **Ildikó Szántó**[4], **Judit Nemes**[5], **Emil Segatto**[1]*

1 Department of Oral and Maxillofacial Surgery, Faculty of Medicine, University of Szeged, Szeged, Hungary, 2 Segatto Dent'art Studio, Budapest, Hungary, 3 Department of Paediatric Dentistry and Orthodontics, Faculty of Dentistry, Semmelweis University, Budapest, Hungary, 4 Department of Paediatric and Adolescent Dentistry, Dentistry Program, Medical School, University of Pécs, Pécs, Hungary, 5 Department of Pediatric and Preventive Dentistry, Faculty of Dentistry, University of Debrecen, Debrecen, Hungary

* segatto.emil@gmail.com

**Data Availability Statement:** The raw dataset is hosted at FigShare (DOI: 10.6084/m9.figshare.

## Abstract

### Background

How dental education influences students' dental and dentofacial esthetic perception has been studied for some time, given the importance of esthetics in dentistry. However, no study before has studied this question in a large sample of students from all grades of dental school. This study sought to fill that gap. The aim was to assess if students' dentofacial esthetic autoperception and heteroperception are associated with their actual stage of studies (grade) and if autoperception has any effect on heteroperception.

### Methods

Between October 2018 and August 2019, a questionnaire was distributed to 919 dental students of all 5 grades of dental school at all four dental schools in Hungary. The questionnaire consisted of the following parts (see also the supplementary material): 1. Demographic data (3 items), Self-Esthetics I (11 multiple-choice items regarding the respondents' perception of their own dentofacial esthetics), Self-Esthetics II (6 Likert-type items regarding the respondents' perception of their own dentofacial esthetics), and Image rating (10 items, 5 images each, of which the respondents have to choose the one they find the most attractive). Both the self-esthetics and the photo rating items were aimed at the assessment of mini- and microesthetic features.

### Results

The response rate was 93.7% (861 students). The self-perception of the respondents was highly favorable, regardless of grade or gender. Grade and heteroperception were significantly associated regarding maxillary midline shift (p < 0.01) and the relative visibility of the arches behind the lips (p < 0.01). Detailed analysis showed a characteristic pattern of preference changes across grades for both esthetic aspects. The third year of studies appeared to be a dividing line in both cases, after which a real preference order was established. Association between autoperception and heteroperception could not be verified for statistical reasons.

11352218). The questionnaire and photo series are provided as supplementary material.

**Funding:** The author(s) received no specific funding for this work.

**Competing interests:** The authors have declared that no competing interests exist.

## Conclusion

Our findings corroborate the results of most previous studies regarding the effect of dental education on the dentofacial esthetic perception of students. We have shown that the effect can be demonstrated on the grade level, which we attribute to the specific curricular contents. We found no gender effect, which, in the light of the literature, suggests that the gender effect in dentofacial esthetic perception is highly culture dependent. The results allow no conclusion regarding the relation between autoperception and heteroperception.

## Introduction

Most contemporary societies put heavy emphasis on esthetic personal appearance. It is well documented that dentofacial esthetics has a profound influence on one's social perception [1], and on self-perception as well [2–4]. For some time now, esthetics has been a central issue in dentistry, as patients put increasing emphasis on this aspect—sometimes prioritizing it over functional considerations [5, 6]. This "esthetic turn" in patients' expectations prompted dental professionals to investigate what counts as esthetic in dentistry, both from the perspectives of the patient and the dentist.

A specific question that has been investigated in several studies since the 1980s is whether dental education (the process of becoming a professional) has a significant effect on one's perception of dental esthetics [7–12].

While the methodologies and sample sizes of earlier studies dealing with the esthetic perception of dental students were mostly proper, no study has ever considered all grades of the dental school comprehensively. Armalaite and colleagues studied only 4th- and 5th- year students [12], the student sample of Tufekci and colleagues consisted of 1st- and 3rd- year students, while other studies did not even differentiate between grades, they treated dental students as a homogeneous group instead.

Another issue that has been brought up by the cited studies is that of gender. The literature is quite divided as to whether gender is a factor in dentofacial esthetic perception in the context of becoming a dental professional.

In the present questionnaire- based, cross-sectional study, we sought to give a description of the dynamics of dental esthetic perception in the process of dental education, covering all five grades of the dental school in all four dental schools in Hungary. We sought to examine in this large student sample if there is association between the stage of studies (grade) and a) one's esthetic self- perception and b) one's esthetic preferences regarding various mini- and microesthetic features, and c) if satisfaction with one's dentofacial esthetics influences dentofacial esthetic preferences. We also sought to study if gender plays a significant role in any of these. As null hypotheses, we hypothesized that a) neither grade nor gender would have a significant association with the students' self-perception and b) esthetic preferences and that c) self-perception would not have an effect on esthetic preferences.

## Methods

### Participants and sampling

All five grades of all the four dental faculties of Hungary (associated with the universities in Szeged, Debrecen, Pécs and Budapest) were involved in the study. After having obtained the deans' approval, printed questionnaires were distributed among the faculties. Each faculty received questionnaires the total number of their Hungarian-speaking students in each grade

at the time +10% (to cover lost, damaged, etc. copies). Altogether 1011 questionnaires were distributed for 919 students. The final sample size was determined by the number of non-responders (see below). The authors administered the questionnaires personally to ensure that the instructions and explanations would be the same at all sites and in each grade. 30 minutes were allocated for answering the questions (including the rating of the photos). The questionnaires were anonymous, and participation was voluntary. Sampling took place between October 2018 and August 2019. The study protocol and the applied instrument were approved by the Regional Ethics Committee for Research in Human Medical Biology at the University of Szeged (No. 178/2018-SZTE). Written informed consent was not required. The students were free not to participate or quit at any time.

## The instrument

Several instruments exist that are related to dentofacial esthetics in some way. Traditional indices, such as the *Dental Esthetic Index* (DAI) or the *Index of Orthodontic Treatment Need* (IOTN) are valid and reliable, but they were developed for older adults [13], and the assessment of subjective satisfaction is not among the primary aims of these instruments [4]. For these reasons, we could not use them for our study. Other indices, like *Oral Health Related Quality of Life* (OHRQoL) do concentrate on subjective factors, but predominantly as determined by oral health in general [14], with esthetics only as a marginal factor. The *Psychosocial Impact of Dental Esthetics Questionnaire* (PIDAQ) and the *Orthognathic Quality of Life Questionnaire* (OQLQ) may be the closest to what we needed for the present study, but still, we found that some combination of the existing questionnaires would be the optimal.

It was for this reason that we developed our own instrument, by combining items from already existing ones [2, 3, 15–17]. The instrument got the working name *Dentofacial Esthetics Instrument for Dental Students*, was developed first in the Hungarian language (an authenticated English translation exists but has not been tested), and it consists of the following parts (see also the supplementary material): 1. Demographic data (3 items), Self-Esthetics I (11 multiple choice items regarding the respondents' perception of their own dentofacial esthetics), Self-Esthetics II (6 5-grade Likert-type items regarding the respondents' perception of their own dentofacial esthetics), and Image rating (10 items, 5 images each, of which the respondents have to choose the one they find the most attractive). Both the self-esthetics and the photo rating items were aimed at the assessment of mini- and microesthetic features, as defined by Sarver [18]. Those features were chosen that are considered to influence the esthetic perception of a smile the most [19, 20].

The smile photos for rating we prepared ourselves. The choice of the model was based on the literature [10, 21, 22]: a clinically normal smile with normal occlusion and features that are generally perceived as esthetic. The image was a standard frontal view showing the anterior teeth, the surrounding gingival tissues and the lips. The image was cropped to remove the chin, nose and cheeks to exclude the confounding effect of macroesthetic features. The photos were taken from a 1.5 m distance, with a Nikon D7000 camera equipped with a Nikon 105mm F 2.8G VR AF-S ED.IF Nikkor objective. The model was standing while being photographed. The photo shooting session took place on a sunny day, in a room amply and evenly lit by natural light, at noontime.

For all esthetic features, the original (unmodified) photo and four modified versions of it were used, so all series (as defined by one feature) consisted of 5 photos (Fig 1). Modifications were always made to the unmodified image. For the modifications, Adobe Photoshop CC 2015 (Adobe Systems, USA) was used. The resulting 50 photos were arranged in an album in a way that items from the same series were always shown on the same page, in random order. Life-

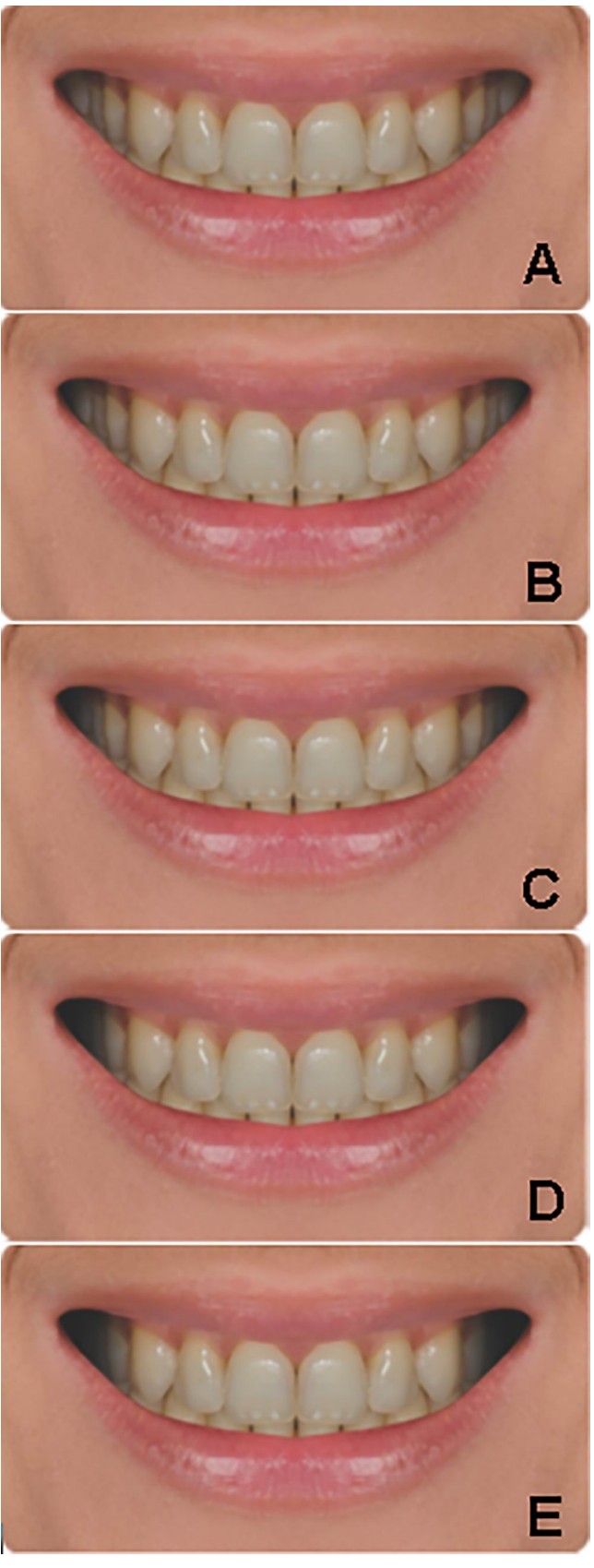

**Fig 1. A sample series of images from the album used for the study.** The modified feature is the width of the buccal corridor. A: the unmodified image. B-E: the buccal corridor widens by 1 mm steps up to + 4 mm (compared to the unmodified image).

size photos were shown. The respondents were not told which the unmodified variant was. All image series with explanation are provided as supplementary material. The modifications are summarized in Table 1. Please note that the questionnaire had not been formally evaluated before this study.

## Statistical analysis

Statistical analysis was carried out by an independent evaluator (see Acknowledgements). The evaluator received the results in a coded format and was told what analyses to carry out according to which codes. The evaluator was blinded to the meaning of the codes.

The results were analyzed in SPSS 22.0 (IBM, USA). Continuous variables were characterized by means with standard deviations and medians, categorical variables were described with the number of observed cases and frequencies expressed in percentages. For hypothesis testing, linear regression analysis, the chi-square test and the Kruskal-Wallis test were used, as appropriate. For the analyses, the multiple-choice items were treated as categorical variables and the Likert-type responses were treated as continuous variables (as they express degree rather than discrete options). The level of significance was $p = 0.05$, unless otherwise indicated.

## Results

### Participants

Of the 919 students, 861 (93.7%) responded. Response willingness to the individual items was also quite good, we managed to obtain 738 to 861 responses to each item. Items with missing

**Table 1. Photo rating: The modified mini- and microesthetic features.** See also supplementary material.

| Series No. | Item No. | Feature | Description |
|---|---|---|---|
| 1 | 4.1. | Smile arc (SA) | The relation of the maxillary incisal edges to the lower lip. From markedly convex to markedly inverted. Arc modified by tilting the maxillary arch in 10° steps upward and downward. |
| 2 | 4.2 | Gingival smile (GS) | The degree of the visibility of the upper gingiva is modified by shifting the arches downward. |
| 3 | 4.3. | Length of canines and lateral incisors (LCLI) | The length of the canines and lateral incisors is modified, in varying combinations. |
| 4 | 4.4. | Width of buccal corridor (WBC) | The width of the buccal corridor is increased, up to +4 mm as compared to the original. |
| 5 | 4.5 | Vertical position of the canine cusps (VPCC) | The position of the canine cusps is shifted above and below the arch level. |
| 6 | 4.6 | The zenith of the front teeth (ZFT) | Horizontal, upward arching and downward arching variations. |
| 7 | 4.7 | The length of frontal interdental papillae (LFIP) | The length of the interdental papillae is modified a) between the central and lateral incisors and b) between the lateral incisors and canines. |
| 8 | 4.8 | Midline shift (MS) | The midline, defined as the vertical line between the central incisors, is shifted to right and left. |
| 9 | 4.9 | Position of the commissures (PC) | The position of the commissures is shifted downward. |
| 10 | 4.10 | The relative visibility of the arches (RVA) | The degree of the relative visibility of the arches is modified by moving them vertically behind the lips. |

**Table 2. The number of students and distribution of genders by grade.**

| Grade | N | Gender (N(%)) |
|---|---|---|
| 1st | 187 | M: 71 (38%) |
| | | F: 116 (62%) |
| 2nd | 184 | M: 61 (33%) |
| | | F: 123 (67%) |
| 3rd | 137 | M: 41 (30%) |
| | | F: 96 (70%) |
| 4th | 169 | M: 62 (37%) |
| | | F: 107 (63%) |
| 5th | 184 | M: 66 (36%) |
| | | F: 118 (64%) |

responses were: 1.2 Age (N = 842); 2.8 Upper central midline (N = 763); 2.9 Lower central midline (N = 742); 2.10 Upper and lower incisal midline alignment (N = 738); 2.11 Gummy smile (N = 774). The mean age of the 861 respondents was 22.34 ± 4.2 years. The number of students and distribution of genders in each grade are given Table 2.

## Esthetic perception of own smile

As the chi-square test did not indicate a significant association between the responses to the multiple choice items (2.1–2.11) and grade, and neither did we find a significant association with gender (entered as a control variable), we concluded that grade and gender made no significant difference on how esthetic students perceived their own smile according to the parameters we asked them about. Thus, to characterize students' self-perception, we analyzed the answers to these items as a whole, instead of breaking the analysis down to grades.

The percentages show that the students had a generally positive perception of their own smile. 210 respondents (24.4%) indicated that they had tried to avoid smiling because of the condition of their teeth at some point in their life or that it was a problem for them at the time of sampling. 116 students (13.5%) indicated that they smile with closed mouth while being photographed. 556 students (64.6%) said that they had never looked at somebody else wishing that their smile would be like the smile of that person. 515 students (59.8%) responded with "no" to the question as to whether they wished to change their smile in any way. The vast majority of the respondents, 740 students (85.9%) were absolutely or partially satisfied with the color of their teeth. Only 121 respondents (14.1%) indicated complete dissatisfaction. A similarly high number of respondents, 688 students (79.9%) found that the shape and size of their teeth were in harmony. Crowding of the teeth was a problem to 365 students (42.4%), and 90 of them indicated that they had crowded teeth in both arches. 710 (82.5%) students indicated that their interincisal line fell exactly in the midline of their face. 587 (68.2%) students found that their lower interincisal line fell exactly in the midline of their face. 508 (59%) students indicated that the upper and lower interincisal lines fell in the same line. It must be noted in connection with the latter 3 items that >10% of all respondents indicated that they had never observed these features of their smile. Finally, 586 (68.1%) students said no when asked whether their upper gum was visible while smiling. 208 (24.2%) students said that their gum was visible while smiling, and 67 (7.8%) students said that they had never observed/considered this.

As for the Likert-type self-evaluation (items 3.1–3.6), we used linear regression analysis to determine if grade or gender had significant effect on the respondents' choices. The effect of grade and gender were analyzed in the same model. Similarly to the multiple choice items, no

**Table 3. Results of the Likert-type items.** The respondents had to indicate how much they agreed with the given statements on a 1 to 5 scale where 1 meant not at all and 5 meant absolutely.

| Item | N | Mean | SD | Median | Mode | Frequency of mode | Min. | Max. |
|---|---|---|---|---|---|---|---|---|
| 3.1. How satisfied are you with the shape of your teeth? | 861 | 4.34 | 0.76 | 4.00 | 5 | 48.32% | 0.00 | 5.00 |
| 3.2. How satisfied are you with the size of your teeth? | 861 | 4.43 | 0.83 | 5.00 | 5 | 56.80% | 0.00 | 5.00 |
| 3.3. How satisfied are you with the orderliness of your teeth? | 861 | 3.86 | 0.96 | 4.00 | 4 | 39.61% | 0.00 | 5.00 |
| 3.4. How satisfied are you with the whiteness of your teeth? | 861 | 3.66 | 0.94 | 4.00 | 4 | 43.32% | 0.00 | 5.00 |
| 3.5. How satisfied are you with the harmony between your teeth and lips? | 861 | 4.36 | 0.87 | 5.00 | 5 | 55.17% | 0.00 | 5.00 |
| 3.6. How satisfied are you with the aesthetics of your smile in general? | 861 | 4.07 | 0.87 | 4.00 | 4 | 45.53% | 0.00 | 5.00 |

significant effect was found for any of the items, and thus the sample was analyzed descriptively and as a whole. The results are summarized in Table 3. The students were the most satisfied with the size of their teeth and the least satisfied with the whiteness (color) of their teeth.

## Photo rating

The chi-square test indicated significant association between grade and items 4.8 (midline shift, $p < 0.01$) and 4.10 (the relative visibility of the arches, $p < 0.01$). No significant association was found with gender for any of the photo series. Figs 2 and 3 show the distribution of preferences across the five grades.

As for the midline shift (Fig 2), it was a common finding for all grades that the 2 mm right shift was the least preferred modification. Regarding the 2 mm left shift, an interesting pattern was found: in the first grade, it was the second most preferred modification, almost exactly as preferred as the most highly rated 1 mm left shift (with only 1% difference). In the second grade, however, it ranked only fourth, but in the third grade, it became the most preferred variation. In the fourth grade, it fell back to the third place, and in the fifth grade it ranked third too. To sum it up, there was a significant variation in the preference for this specific modification across the grades. Another finding to mention is that the unmodified image was the most preferred only in the fourth grade, in the rest of the grades, it ranked third. In the first, second and fifth grades, the 1 mm left shift was the most preferred variation. In general, the preferences were quite evenly distributed across the variations, with not much difference between the variations. The only exception seems to be the fifth grade, where the differences between the preferences for each variation were the largest, resulting in what can be considered a real rank order. Furthermore, the third year appears to be a turning point, where preferences

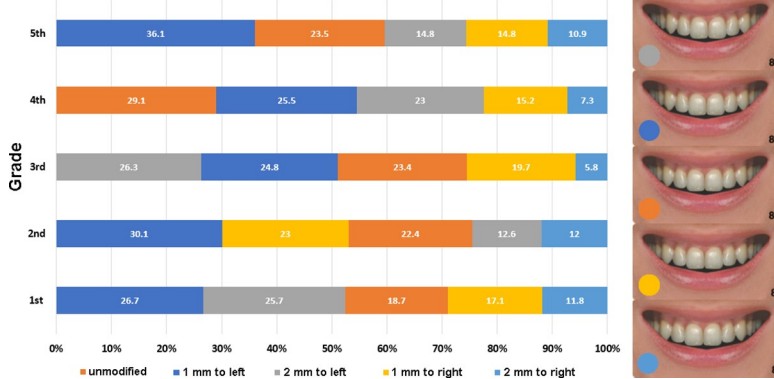

**Fig 2. Distribution of preferences in the five grades for midline shift.** The vertical line between the central incisors was shifted to the left and to the right in 1 mm steps (percentages, N = 861).

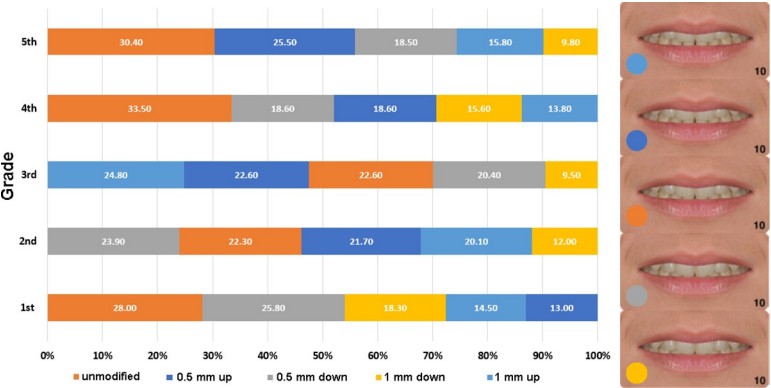

**Fig 3. Distribution of preferences in the five grades for the relative visibility of the arches.** The arches were moved upward and downward behind the lips in 0.5 mm steps (percentages, N = 861).

stabilize, and from the fourth year on, no real variation is seen in the preference order. Comparing the first and fifth grades, it is interesting to see that the preferences of the two grades are almost the same, with the single exception of the 2 mm to left variation, which is almost as preferred as the first-ranking choice in the first year, but clearly falls back to the third place by the fifth year.

As for the relative visibility of the arches (modified by moving the arches up and down behind the lips, Fig 3), the variability of preferences across the grades is obvious at first sight. The unmodified variation turned to be the most preferred in 3 of the 5 grades, but it never ranked lower than 3rd (in the third grade). The least preferred variation, almost regardless of grade, was the 1 mm downward shift (where the least of the lower arch is visible). Preferences for the 1 mm up and 0.5 mm down variations varied the most across the grades. Similarly to what was observed in connection with the midline shift, the third grade appears to be a turning point, from where on, the preferences are distributed across the variations less evenly, and a real rank order seems to develop. Unlike in the case of the midline shift, the preference order in the fifth grade was markedly different from that in the first grade, except for the unmodified variation being the most preferred one in both grades.

Regarding the rest of the photo series, we did not find significant association with grade, that is, the sample showed approximately the same preferences, regardless of grade. In Table 4, however, we provide the most and least preferred variations by grade, to give a characterization of the preferences of the sample in general.

**Table 4. The most and the least preferred variations by grade for the photo series where no significant association with grade was found (N = 861).** The conventions are the same as in Table 1.

| Feature | Most preferred (%) | Least preferred (%) |
|---|---|---|
| SA | Inverse 1 (55%) | Concave 1 (0.6%) |
| GS | Arches 2 mm up (44%) | Arches 2 mm down (2.7%) |
| LCLI | Canines 1 mm shorter (29%) | Canines and lateral incisors 1 mm longer (11.5%) |
| WBC | Unmodified (23.5%) | Buccal corridor 2 mm wider (16%) |
| VPCC | Cusps 1 mm above arch level (24.8%) | Unmodified (15.8%) |
| ZFT | Horizontal zenith (35.5%) | Upward arching (8.73%) |
| LFIP | Unmodified (31.2%) | All papillae shorter (5.74%) |
| PC | Unmodified (36%) | Commissures 3 mm downward (12.35%) |

### Self-perception and photo rating

Significant associations were found between items 3.1 and 4.10 for gender ($p < 0.05$), 3.2. and 4.10 for gender and grade ($p < 0.01$ and $p < 0.01$, respectively), and between 3.5 and 4.4 for gender ($p < 0.05$). An in-depth crosstable analysis showed, however, that these were only apparent effects stemming from the distorting effect of the extremely low number of students with low self-evaluation scores. We concluded that the data did not support the existence of significant association between dentofacial autoperception and heteroperception.

## Discussion

The perception of dentofacial esthetics by dental students and dentists has been studied since the 1980s. Brisman [7] compared patients, dental students and dentists regarding their esthetic preferences in connection with maxillary central incisors. He found that the three groups significantly differed in their preferences (indicating an effect of education), and that gender was not a factor. Phillips et al. [8], studying orthodontics residents, dental students and non-dental university students came to a similar conclusion regarding the effect of education. Tufekci et al. [9] came to the same conclusion in their study of laypeople, orthodontic patients, and first- and third-year dental students. Finally, Kokich et al. [10] demonstrated that being an orthodontist does make one more critical about a number of dental esthetic features than general dentists or laypersons. Interestingly enough, one study found no difference at all between dentists and laypersons in this respect, but this can be probably put down to the unusually small sample size of 10 respondents per group [11]. In summary, there seems to be a consensus in the literature, regardless of geographical area, that studying dentistry does make a difference in one's perception of various dental esthetic features. There is less consensus as to whether the gender of the student or dentist makes a difference. As mentioned before, Brisman [7], working in the New York area, found no gender effect. In contrast, Armalaite [12], studying a larger sample of Lithuanian dental students, found a definite and significant difference between males and females in the evaluation of a number of dental esthetic features from gingival smile through dental crowding. They drew the conclusion that women were "more critical" regarding these features. Finally, Alhammadi and colleagues [23] studied 408 Saudi Arabian dental students, and came to the conclusion that "male dental students have a better perception of facial and dental esthetics." It is not clear if the remarkable difference between the results of these studies is due to the different geographical locations, the difference between the studied features or something else.

The general aim of the present study was to give a comprehensive, cross-sectional characterization of a large sample of dental students from all grades of dental school in terms of their dental esthetic (self-) perception, also considering gender as a factor. The reason for this was that no study before has examined all grades together from the same cultural and geographical region. We have shown that the effect of dental education can be demonstrated on the grade level, however, no gender effect was found and no association between autoperception and heteroperception could be reliably demonstrated. When interpreting our results, it must be taken into consideration that the questionnaire was our own development and had not been formally evaluated before the study, which is definitely a limitation.

Specifically, we sought to answer four research questions. First, we asked if there is association between the stage of studies (grade) and dentofacial esthetic self- perception. We asked this question because we assumed that dentofacial self-perception could influence dentofacial preferences (hence the third research question). This we assessed with multiple-choice questions. Had we found such associations, we should have taken them into consideration when analyzing the association between grade and esthetic preference. We found no such

associations, though, and neither was the respondents' gender significantly associated with self-perception. It seems that the 861 respondents formed a more or less homogeneous, and generally satisfied, group regarding dentofacial esthetic self-perception. While there were lower-scoring items (like the whiteness of the teeth), none of the studied aspects scored an average below 3 on a scale of 5. It is difficult to compare these results to any other published result, as few studies before asked these questions specifically in connection with dental students. The lack of gender effect was somewhat surprising, as literature (of non-dental) students suggests that female students tend to be more critical about their body [24, 25]. It is to be taken into consideration, though, that these studies mostly concentrated on body image as related to body mass. Could it be that one's dentofacial body image is an exception? Or is it simply that dental students comprise a special group in this respect? The 2016 study of Strajnic and colleagues [26] failed to find gender difference in self-perception and satisfaction with dental appearance in patients (i.e. laypersons), which supports the former hypothesis. However, this question has been rarely discussed in the literature and definitely needs further investigation.

The second question was if there is association between grade and esthetic preferences regarding various mini- and microesthetic features. To assess this, we used digitally manipulated images. Significant association was found between grade and a) midline shift and b) the relative visibility of the arches, indicating that the preferences regarding these features showed the highest variability across the grades. Gender did not turn out to be a significant factor in this analysis either, so the results were analyzed by grade alone.

The importance of the maxillary midline in dentofacial esthetics is well documented. Johnston and colleagues [27] studied the esthetic judgements of orthodontists and laypersons and found that the greater the shift, the more unattractive the given smile is found. This was true both for the professionals and the laypersons, regardless of gender, and reached significance if the shift was >2 mm (in either direction). In the study of Armalaite et al. [12], shifted maxillary midline was classified by both male and female (4th- and 5th- year) dental students as unacceptable. Alhammadi and co-workers report that the dental students in their study noticed as small a shift as 1 mm [23]. It seems, thus, that the position of the maxillary midline is a sensitive esthetic issue, so it came as no surprise that in our study it was one of the parameters significantly associated with the stage of studies. It is an intriguing finding that in 3 of the 5 grades, the 1 mm left- shifted variation was the most preferred, and even in the remaining two grades (the 3rd and the 4th), it was the second most preferred, only a few per cents behind the most preferred ones. The unmodified variation was the most preferred only in the 4th year. It is difficult to offer any rational explanation other than that the studied population does actually prefer a not precisely centered midline. While studies take zero deviation as the norm, it is safe to assume that preferences in this respect vary across cultures and geographical areas. As for the supposed effect of dental education, Fig 2 clearly shows the difference of preference orders between the grades and that a real rank order with actually meaningful differences is established only by the end of the studies.

The other modified feature to show significant association with grade was the relative visibility of the upper and lower arches. While there was a clear tendency in all grades toward preferring the unmodified variation (it was the most preferred one in three of the five grades), the 1 mm up and 0.5 mm down variations obviously divided the students. For instance, the 1 mm up variation (ranked generally low) was the most preferred in the 3rd grade, but the least preferred in the 4th. What the reason is for this high variability regarding these variations is not clear. The general unpopularity of the 1 mm up variation would be easily explained by the fact that it could remind respondents of edge-to-edge bite, taught as a form of malocclusion from the earliest stages of dental studies. However, it does not explain the high popularity seen in

the 3<sup>rd</sup> grade. If this is a unique result and not an artifact, it probably reflects the fact that students in the Hungarian dental curricula begin to deal with the finer details of dental esthetics after the 3<sup>rd</sup> grade (in the clinical module of the curriculum). This would explain why preference ratios do not reflect a clear rank order before the 4<sup>th</sup> and 5<sup>th</sup> grades. While modification of the midline is a relatively crude modification, the relative visibility of the arches is a finer issue. It is possible that the 0.5 mm steps of modification were too fine for the students to allow a clear-cut order of preference before being able to think consciously about it. However, in lack of published data on this parameter, it is difficult to offer a firm explanation.

Why exactly the perception of these two features showed association with grade might have several reasons, from cultural factors through the exact contents of education regarding dental esthetics. However, the discussion of such background factors is clearly beyond the scope of this paper, especially that we did not gather related data. The results, however, confirm the finding of other studies that there is an association between dental knowledge (expressed as grade in this study) and the esthetic perception of dentofacial features.

As for the items not significantly associated with grade, the following pattern was found: the unmodified variation was the most preferred one in only 3 items of the 8 (the width of the buccal corridor, the length of the frontal interdental papillae and the position of the commissures), and in one item (the vertical position of the canine cusps) it was the least preferred one. The latter finding was not entirely unexpected: our photo model had long canines, which means that in the unmodified variation of the photo, the canines extended below the incisal edges of the central incisors. Li and colleagues, working with similar frontal photos to ours, found that both orthodontic experts and laypersons showed low tolerance regarding the difference between the length of the central incisors and canines and preferred situations when the canines where slightly shorter [28]. Our results corroborate this: shorter canines were preferred over longer ones. As for gingival smile, the choices indicated a marked preference for the invisibility of the gingiva, the variation with the least visible gingiva being the most preferred one. At first sight, our results seem to agree with the literature. The student sample of Armalaite et al. [12] rated gingival smile "esthetically unacceptable", and there appears to be a general agreement in the literature that gingival smile is a medical condition, as treatments are proposed [29]. It must be added, though, that this extremely negative evaluation is not uniform across cultures: studies from Brazil [30] and Japan [31] reported that the visibility of the gingiva does not necessarily influence the esthetic judgement of smile. Concerning the smile arc, the slightly inverted arc was the most frequent choice, which, again, is a slight deviation from what is generally considered to be the esthetic norm (i.e. a slightly convex arc that contours the lower lip) [32]. These can be regarded as esthetic preferences that were generally characteristic of the entire sample, regardless of grade and gender.

Finally, we asked if satisfaction with one's dentofacial esthetics influences dentofacial esthetic preferences. In this respect, we found significant associations between satisfaction with the shape of one's teeth and the relative visibility of the arches in the photo series; satisfaction with the size of one's teeth and the relative visibility of the arches in the photo series; and satisfaction with the harmony between one's teeth and lips and the width of the buccal corridor in the photo series. The detailed crosstable analysis indicated that the observed significance in these cases was statistical artifact, and thus we concluded that our data did not allow the conclusion that dentofacial autoperception influences dentofacial heteroperception. This does not necessarily mean that such an effect does not exist, though. Even if our sample was large enough to allow conclusions, our respondents were predominantly satisfied with their dentofacial esthetics, which confounded the analysis. Further studies are definitely needed, especially that, to our knowledge, this study has been the first to ask this question.

## Conclusions

To our knowledge, our study has been the first to study the dentofacial esthetic auto- and het-eroperception of dental students at all grades of dental school, in a large and culturally homo-geneous sample. Our findings corroborate the results of most previous studies regarding the effect of dental education on the dentofacial esthetic perception of students. We have shown that the effect can be demonstrated on the grade level, which we attribute to the specific curric-ular contents. In this respect we suggest that the study of how specific curricular contents influence the dentofacial esthetic perception of students is a promising direction for future research. We found no gender effect, which, in the light of the literature, suggests that the gen-der effect in dentofacial esthetic perception is highly culture dependent. Finally, we failed to confirm association between autoperception and heteroperception, but as the analysis was confounded by the uneven distribution of the data, we cannot conclude that no such associa-tion exists.

## Supporting information

**S1 Data. Dentofacial aesthetics instrument for dental students.**
(PDF)

**S2 Data.**
(PDF)

## Acknowledgments

The authors would like to express their gratitude to Dr. Gábor Braunitzer of dicomLAB Dental for his selfless help with data management and analysis.

## Author Contributions

**Conceptualization:** Lívia Romsics, Angyalka Segatto, Kristóf Boa, Ildikó Szántó, Judit Nemes, Emil Segatto.

**Data curation:** Noémi Rózsa, Judit Nemes.

**Investigation:** Lívia Romsics, Angyalka Segatto, Kristóf Boa, Roland Becsei, Noémi Rózsa, Ildikó Szántó, Judit Nemes.

**Methodology:** Roland Becsei, Noémi Rózsa, Judit Nemes.

**Project administration:** Angyalka Segatto.

**Supervision:** Angyalka Segatto, Emil Segatto.

**Writing – original draft:** Lívia Romsics, Angyalka Segatto, Roland Becsei, Emil Segatto.

**Writing – review & editing:** Kristóf Boa, Noémi Rózsa, Ildikó Szántó, Judit Nemes.

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
