## [Decision Letter · Decision Letter 0]

3 Jan 2020

PONE-D-19-34253

Dentofacial Mini- and Microesthetics as Perceived by Dental Students: A Cross-Sectional Multi-Site Study

PLOS ONE

Dear Dr Segatto,

Thank you for submitting your manuscript to PLOS ONE. After careful consideration, we feel that it has merit but does not fully meet PLOS ONE’s publication criteria as it currently stands. Therefore, we invite you to submit a revised version of the manuscript that addresses the points raised during the review process.

We would appreciate receiving your revised manuscript by Feb 17 2020 11:59PM. To enhance the reproducibility of your results, we recommend that if applicable you deposit your laboratory protocols in protocols.io, where a protocol can be assigned its own identifier (DOI) such that it can be cited independently in the future. For instructions see: http://journals.plos.org/plosone/s/submission-guidelines#loc-laboratory-protocols

We look forward to receiving your revised manuscript.

Kind regards,

Wen-Jun Tu

Academic Editor

PLOS ONE

Journal Requirements:

3. Please remove your figures from within your manuscript file, leaving only the individual TIFF/EPS image files, uploaded separately.  These will be automatically included in the reviewers’ PDF.

Reviewers' comments:

Reviewer's Responses to Questions

**Comments to the Author**

1. Is the manuscript technically sound, and do the data support the conclusions?

Reviewer #1: No

2. Has the statistical analysis been performed appropriately and rigorously? 

Reviewer #1: No

3. Have the authors made all data underlying the findings in their manuscript fully available?

Reviewer #1: Yes

4. Is the manuscript presented in an intelligible fashion and written in standard English?

Reviewer #1: No

5. Review Comments to the Author

Reviewer #1: Comments

The methodological conditions related to cross-sectional studies are not applied: STROBE Statement — Checklist of items that should be included in reports of cross-sectional studies

Abstract

Methods

Specify the year of the study

Please provide in the abstract an informative and balanced summary of what was done and what was found

Please systematically replace (χ2 =32.97, df= 16, p < 0.01 with (χ2 =32.97, df= 16, p < 0.01

Association between autoperception and heteroperception could not be verified for statistical reasons: WHY ? It is a major point to explain an discuss in the text

62.Our findings corroborate the results of most previous studies regarding the effect of dental education on the dentofacial esthetic perception of students. There is a contradiction with line 42 : However, no study before has studied this question in a large sample of students from all grades of dental school.

Introduction

It must be shortened. Lots of points can be part of the discussion

Explain the scientific background and rationale for the investigation being reported

Be clearer about the objectives of the study

Materials and Methods

Please respect STROBE Statement Checklist of items

Likert scale. It is always recommended to choose a scale composed of an odd number of degrees: 3, 5, 7, in order to allow the respondent not to position themselves. Why did you offer 6 items?

How did you build the questionnaire? Has it been evaluated?

You write: The instrument got the working name Dentofacial Esthetics Instrument for Dental Students, was developed first in the Hungarian language. Question: Is this index not referenced (line 153-154)

Line 143-149: It is Discussion if necessary

Table 3. Results of the Likert-type items. The respondents had to indicate how much they

Line 173-180: Could you just explain the difference between clinical cases. For a non-specialist, it is really complicated to understand the meaning of these photos.

χ2 = 32.97, df = 16: useless

Table 2. Demographic information by grade. This presentation format is of little interest, especially since the S.Ds seem to indicate that there would be no difference between the grades. To review

Statistics: The question arises. Why not have evaluated the association between autoperception and heteroperception could not be verified for statistical reasons:

Start analyzing Likert scale data with descriptive statistics. While it may be tempting, resist the urge to take the numerical responses and calculate an average. With Likert scale data, the best measure to use is the most common response. This will make the results of the survey much easier to interpret by the analyst and the audience to which it is addressed. You can also represent the distribution of responses (percentages that agree, disagree, etc.) on a graph, such as a bar graph, with a bar for each response category.

Discussion

The reason for this was that no study before has examined all grades together from the same cultural and geographical region. It contradicts what you write in the introduction

Line 358-360: To withdraw

Please strictly follow the strobe statements guidelines

6. PLOS authors have the option to publish the peer review history of their article (what does this mean?). If published, this will include your full peer review and any attached files.

Reviewer #1: No

---

## [Author Response · Author response to Decision Letter 0]

5 Feb 2020

[We recommend the attached file instead]

RESPONSES TO THE REVIEWER

Reviewer #1: Comments

The methodological conditions related to cross-sectional studies are not applied: STROBE Statement — Checklist of items that should be included in reports of cross-sectional studies

R: Thank you for this important observation, we have revised the manuscript accordingly. We would like to note, though, that the STROBE guidelines determine the desired components of a cross-sectional report by main sections, not their mandatory order within the section (even if the checklist items are numbered with ordinals). We have attached a filled checklist with page numbers. 

Abstract

Methods

Specify the year of the study

R: We have specified the years (period). 

Please provide in the abstract an informative and balanced summary of what was done and what was found

R: This is an actual item from the STROBE checklist (1b). We have tried to find the reason why the Reviewer thinks that our abstract does not fit this (i.e. why it is uninformative or not balanced), but we came to the conclusion that without further clues, we cannot make it more informative and balanced. It does what an abstract is supposed to do: it summarizes what happened, why and what conclusions were reached. Any closer suggestions are welcome. 

Please systematically replace (χ2 =32.97, df= 16, p < 0.01 with (χ2 =32.97, df= 16, p < 0.01

R: This is not entirely clear; however, we later found that the Reviewer does not find it appropriate that besides the significance values, the chi-squared statistic and the degrees of freedom are also provided (as per the APA guidelines). While uncertain about whether the reviewer really meant this, we modified the manuscript by deleting the values deemed inappropriate. 

Association between autoperception and heteroperception could not be verified for statistical reasons: WHY? It is a major point to explain an discuss in the text

R: It is clearly stated that the uniformness of self-satisfaction responses prevented us from this. In other words, the characteristics of the sample did not make it possible. 

62.Our findings corroborate the results of most previous studies regarding the effect of dental education on the dentofacial esthetic perception of students. There is a contradiction with line 42 : However, no study before has studied this question in a large sample of students from all grades of dental school.

R: This is only an apparent contradiction. The fact that

a.) no study before has studied this question in a large sample of students from all grades of dental school does not mean that

b.) this question has not been studied in student samples at all. 

In fact, we cite smaller, less comprehensive studies throughout the manuscript. Therefore, what the statement

“Our findings corroborate the results of most previous studies regarding the effect of dental education on the dentofacial esthetic perception of students.”

means is merely that this study (of a larger sample size) corroborates the results of smaller ones regarding the same question. 

Introduction

It must be shortened. Lots of points can be part of the discussion

R: We have shortened the section as requested, from 785 words to 376 words. Some parts have been moved to the Discussion. 

Explain the scientific background and rationale for the investigation being reported

R: This is another item from the STROBE checklist. With all due respect, we find that the Introduction does just that. First, we state why dentofacial esthetics is important (with references), then we go on to summarize the most important findings of previous studies (background). This is followed by the exposition of the problem:

“There seems to be a consensus in the literature, regardless of geographical area, that studying dentistry does make a difference in one’s perception of various dental esthetic features. There is less consensus as to whether the gender of the student or dentist makes a difference. (…) While the methodologies and sample sizes of earlier studies dealing with the esthetic perception of dental students were mostly proper, no study has ever considered all grades of the dental school comprehensively.” 

- which is the rationale (i.e. some conclusions have been drawn, some questions are debated, but no study before has worked with a sample size and composition that allows stronger conclusions. That is why we wished to study a number of questions in a large and comprehensive sample).

Again, we are happy to put it in another way, if we get any closer direction. This way, we cannot tell more than that the Reviewer, for some reason, does not like the way the background and rationale are explained, or does not find the explanation satisfactory. 

Be clearer about the objectives of the study

R: The objectives are given as follows, quote:

“We sought to examine in a large student sample if there is association between the stage of studies (grade) and a) one’s esthetic self- perception and b) one’s esthetic preferences regarding various mini- and microesthetic features, and c) if satisfaction with one’s dentofacial esthetics influences dentofacial esthetic preferences. We also sought to study if gender plays a significant role in any of these.”

This is a point-by-point summary of the objectives, almost like a list (not in a list format, though). We sought to find answers to a)- c), plus the gender perspective. We do not see how it could be any clearer. Again, any suggestions are welcome. 

Materials and Methods

Please respect STROBE Statement Checklist of items

R: See our response to the first comment. 

Likert scale. It is always recommended to choose a scale composed of an odd number of degrees: 3, 5, 7, in order to allow the respondent not to position themselves. Why did you offer 6 items?

R: This is a misunderstanding. 6 does not mean 6 degrees. 6 means that the given part of the questionnaire contained 6 questions with 5-grade Likert-type scales (see Supplement). We have added “5-grade” to clarify this issue. 

How did you build the questionnaire? Has it been evaluated?

R: We combined items from existing questionnaires with the help of the Department of Psychology. We have added this information to the manuscript. Unfortunately, we had not had the chance to formally test the questionnaire before this study. This is a limitation indeed, and this has been added to the manuscript. 

You write: The instrument got the working name Dentofacial Esthetics Instrument for Dental Students, was developed first in the Hungarian language. Question: Is this index not referenced (line 153-154)

R: This instrument is a new instrument of our development which we publish here for the first time; thus, it is not referenced. 

Line 143-149: It is Discussion if necessary

R: We agree with the Reviewer that this information is somewhat misplaced here, being, de facto, background information, we cannot think of any place for it that would be much better. In the Discussion it would be too late to explain why we did not simply use an existing questionnaire. The Introduction (which might be a better place) was too long already, it had to be shortened. Most importantly: how and why we concluded that we would not use an existing questionnaire is a methodological question, and this is a decisive factor. Whether this part is necessary we do not consider questionable: we need to be transparent, the reader has the right to know our reasons. 

Table 3. Results of the Likert-type items. The respondents had to indicate how much they

R: The end of the sentence appears to be missing; we are not sure what the Reviewer meant. Unfortunately, this way, we cannot respond to this point. 

Line 173-180: Could you just explain the difference between clinical cases. For a non-specialist, it is really complicated to understand the meaning of these photos.

R: The modifications are explained in the supplementary material with pictures. The reader is advised to see the supplementary material in the caption for Table 1. Table 1. And the supplementary images together should be enough for the readership interested in this kind of research to understand what we did. By oversimplifying the descriptions, we would risk losing important professional detail, which, in the eyes of the target readership, could make our work look unprofessional. 

χ2 = 32.97, df = 16: useless

R: It seems that the Reviewer thinks that reporting statistics according to the APA guidelines is not appropriate in PLOS One. We deleted all details and left only significance levels. We hope this is what the Reviewer meant. 

Table 2. Demographic information by grade. This presentation format is of little interest, especially since the SDs seem to indicate that there would be no difference between the grades. To review

R: The Reviewer probably means the age column. Indeed, this did not carry much extra information, so we deleted it. Instead, we added the information regarding the entire sample (no breakdown by grades) to the text. 

Statistics: The question arises. Why not have evaluated the association between autoperception and heteroperception could not be verified for statistical reasons:

R: Unfortunately, we could not determine the exact meaning of this comment. One thing is certain: the association could not be verified, given the distorting effect of the extremely low number of students with low self-evaluation scores. In other words, as a dominant majority were highly satisfied with their dentofacial esthetics, which rendered this sample unfit for the analysis of such an association. This information is included in the manuscript. 

Start analyzing Likert scale data with descriptive statistics. While it may be tempting, resist the urge to take the numerical responses and calculate an average. With Likert scale data, the best measure to use is the most common response. This will make the results of the survey much easier to interpret by the analyst and the audience to which it is addressed. You can also represent the distribution of responses (percentages that agree, disagree, etc.) on a graph, such as a bar graph, with a bar for each response category.

R: Thanks for this useful piece of advice. We added mode (the most frequent value and its frequency) as a descriptive statistic to characterize the Likert-type items 

Discussion

The reason for this was that no study before has examined all grades together from the same cultural and geographical region. It contradicts what you write in the introduction

R: With all due respect, there is no contradiction here. In the introduction we write, quote: 

While the methodologies and sample sizes of earlier studies dealing with the esthetic perception of dental students were mostly proper, no study has ever considered all grades of the dental school comprehensively. Armalaite and colleagues studied only 4th- and 5th- year students [18], the student sample of Tufekci and colleagues consisted of 1st- and 3rd- year students, while other studies did not even differentiate between grades, they treated dental students as a homogeneous group instead. 

This exactly means that no study before has examined all grades together from the same cultural-geographical region (i.e. a comprehensive study, involving all grades of the dental school). Armalaite and colleagues (cultural region 1) studied only the last two grades, while Tufekci and colleagues (cultural region 2) only the first three. These are two separate ranges of grade in two separate cultural regions. 

Line 358-360: To withdraw

R: We agreed that the reviewer means that we should delete these lines. We have done so. 

Please strictly follow the strobe statements guidelines

R: See our response to the first comment.

---

## [Decision Letter · Decision Letter 1]

25 Feb 2020

Dentofacial Mini- and Microesthetics as Perceived by Dental Students: A Cross-Sectional Multi-Site Study

PONE-D-19-34253R1

Dear Dr. Segatto,

We are pleased to inform you that your manuscript has been judged scientifically suitable for publication and will be formally accepted for publication once it complies with all outstanding technical requirements.

With kind regards,

Wen-Jun Tu

Academic Editor

PLOS ONE

Additional Editor Comments (optional):

Reviewers' comments:

Reviewer's Responses to Questions

**Comments to the Author**

1. If the authors have adequately addressed your comments raised in a previous round of review and you feel that this manuscript is now acceptable for publication, you may indicate that here to bypass the “Comments to the Author” section, enter your conflict of interest statement in the “Confidential to Editor” section, and submit your "Accept" recommendation.

Reviewer #1: All comments have been addressed

2. Is the manuscript technically sound, and do the data support the conclusions?

Reviewer #1: Yes

3. Has the statistical analysis been performed appropriately and rigorously? 

Reviewer #1: Yes

4. Have the authors made all data underlying the findings in their manuscript fully available?

Reviewer #1: Yes

5. Is the manuscript presented in an intelligible fashion and written in standard English?

Reviewer #1: Yes

6. Review Comments to the Author

Reviewer #1: (No Response)

7. PLOS authors have the option to publish the peer review history of their article (what does this mean?). If published, this will include your full peer review and any attached files.

Reviewer #1: No

---

## [Editor Report · Acceptance letter]

27 Feb 2020

PONE-D-19-34253R1 

Dentofacial Mini- and Microesthetics as Perceived by Dental Students: A Cross-Sectional Multi-Site Study 

Dear Dr. Segatto:

I am pleased to inform you that your manuscript has been deemed suitable for publication in PLOS ONE. Congratulations! Your manuscript is now with our production department. 

With kind regards,

on behalf of

Dr. Wen-Jun Tu 

Academic Editor

PLOS ONE